# Indigenous Cultural Safety in Recognizing and Responding to Family Violence: A Systematic Scoping Review

**DOI:** 10.3390/ijerph192416967

**Published:** 2022-12-17

**Authors:** Ilana Allice, Anita Acai, Ayda Ferdossifard, Christine Wekerle, Melissa Kimber

**Affiliations:** 1Department of Psychiatry and Behavioural Neurosciences, McMaster University, Hamilton, ON L8S 4L8, Canada; 2Offord Centre for Child Studies, McMaster University, Hamilton, ON L8S 4L8, Canada; 3Department of Pediatrics, McMaster University, Hamilton, ON L8S 4L8, Canada; 4Optentia Research Unit, North-West University, Potchefstroom 2520, South Africa

**Keywords:** cultural safety, family violence, Indigenous, aboriginal, systematic scoping review

## Abstract

This systematic scoping review synthesizes the recommended approaches for providing culturally safe family violence interventions to Indigenous peoples in health care and social service settings. A total of 3783 sources were identified through our electronic database searches, hand-searching of Indigenous-focused journals, and backward and forward citation chaining. After screening those sources in duplicate, 28 papers were included for synthesis in June 2020. Forward citation chaining of these 28 included articles in June 2022 identified an additional 304 possible articles for inclusion; following the screening of those 304 articles, an additional 6 were retained in the review. Thus, a total of 34 articles were included for data extraction and narrative synthesis. Initial results were presented to members of the Six Nations of the Grand River Youth Mental Wellness Committee, and their feedback was incorporated into our inductive organization of findings. Our findings represent three thematic areas that reflect key recommendations for health care and social service provision to Indigenous families for whom family violence is a concern: (1) creating the conditions for cultural safety; (2) healing at the individual and community level; and (3) system-level change. These findings demonstrate the need to *center* Indigenous peoples and perspectives in the development and implementation of cultural safety approaches, to acknowledge and address historically contingent causes of past and present family violence including colonization and related state policies, and to transform knowledge and power relationships at the provider, organization, and government level.

## 1. Introduction

Exposure to family violence, which includes intimate partner violence (IPV), child maltreatment (CM), and childhood exposure to intimate partner violence (CEIPV) between parents, can have serious negative short- and long-term consequences on the physical and emotional well-being of people and families [1]. While family violence is a global issue that affects people from all communities, structural and interpersonal forms of racism, including the historical and ongoing impacts of colonization, disproportionately increase the risk for experiencing all forms of family violence for Indigenous peoples [2,3,4,5]. Globally, Indigenous peoples are more likely than their non-Indigenous counterparts to experience all forms of family violence [6,7,8,9,10,11]. For example, in Canada alone, Indigenous peoples are two times more likely than non-Indigenous peoples to report IPV victimization by a current or former partner in the previous five years [12]. First Nations children are still vastly over-represented in the child welfare system [13], and First Nations families are more than four times more likely to be investigated for child abuse or neglect, compared with non-Indigenous Canadians [14]. In addition, Indigenous women have the greatest likelihood of experiencing the most chronic and severe forms of family violence, including childhood physical and sexual abuse, domestic homicide, sexual assault, strangulation, and severe physical IPV [15]. These forms of family violence can result in devastating mental, emotional, and physical health outcomes, including depression, post-traumatic stress disorder, and chronic pain, among others [16,17,18].

Given the health-related correlates and consequences of family violence, it is likely that Indigenous peoples who have these experiences will require health and social services related to this exposure. Optimally, the providers in these systems would be able to recognize that family violence is occurring, support the prevention of recurrence, and assist with reducing associated mental and physical health impairments. However, health care and social service organizations are frequently unsafe, alienating, and intimidating for Indigenous peoples; a lack of trust, as well as frequent stereotyping and racism, may prevent Indigenous peoples from disclosing health-related concerns, including family violence [19]. Evidence also indicates that health care and social service providers are inadequately prepared with the knowledge, skills, and behaviors to respond to disclosures of family violence in their practice encounters [20,21]. Unsurprisingly, Indigenous peoples have expressed a preference to use their familial- and community-based informal support networks when experiencing violence and only use formal providers when reaching a crisis point [22]. Without adequate preparation and training for providers to work with Indigenous peoples experiencing violence, there is considerable risk of providers re-perpetuating structural and interpersonal forms of discrimination and harm within the health care setting. 

Renewed scrutiny of the harms that occur for Indigenous peoples within health care settings, as well as global commitments to enact the United Nations Declaration on the Rights of Indigenous People [23], have together leveraged a surge in political, educational, and organizational efforts to improve the awareness of Indigenous-specific racism in health care encounters and mitigate its impacts [24]. One example of these efforts includes the development, evaluation, and scale-up of the concept of ‘cultural safety’ and related programs, training, and interventions (collectively hereafter referred to as ‘programs’). In general, these programs aim to improve health care providers’ awareness about the prevalence and impacts of racism in health care, reduce the possibility of service-related harms, and improve the quality of care provided to Indigenous peoples [25,26,27,28,29,30]. The term cultural safety was developed in the mid-1990s by Māori nurse and scholar Irihapeti Ramsden [31,32], who described it as an “outcome of nurse and midwifery training that enables safe service to be defined by the service user” Table 3 in [32] (p. 119). Ramsden’s concept of cultural safety grew out of the post-colonial context of Māori health inequalities and was concerned with the transfer of power and the establishment of trust between clinician and client, with the encounter defined by the perspective and experience of the Indigenous person, rather than by the provider [32]. Ramsden also positioned culturally safe care as distinct from cultural sensitivity and cultural competence, in that cultural safety requires providers to reflect on their *own* personal and professional culture and positions of power, with the goal to build trusting relationships that would invite patients to share information about their culture on their own terms, and ultimately to provide treatment that attends to that cultural difference. 

Over the past 25 years, many different terms and definitions have become linked to the broad goals of working across cultural lines and improving health care outcomes and experiences for Indigenous peoples [33]; these terms include cultural competence, cultural sensitivity, culturally appropriate care, cultural humility, trauma-and-violence-informed care, and anti-racist and anti-oppressive approaches, among others. For the purposes of this systematic scoping review, we drew on a guiding definition of cultural safety provided by Curtis and colleagues [34] to identify key components of culturally safe care, while being inclusive of various terminology. Their guiding definition states:


*Cultural safety … requires individual healthcare professionals and healthcare organisations to acknowledge and address their own biases, attitudes, assumptions, stereotypes, prejudices, structures and characteristics that may affect the quality of care provided. In doing so, cultural safety encompasses a critical consciousness where healthcare professionals and healthcare organisations engage in ongoing self-reflection and self-awareness and hold themselves accountable for providing culturally safe care, as defined by the patient and their communities, and as measured through progress towards achieving health equity. Cultural safety requires healthcare professionals and their associated healthcare organisations to influence healthcare to reduce bias and achieve equity within the workforce and working environment.*
(p. 188)

Although calls for cultural safety in health and social service provider training and service provision in the family violence context have begun to appear in specific organizational contexts (e.g., primary care [2]), research that details and consolidates specific information about how to create culturally safe care for Indigenous peoples experiencing family violence is lacking. To address this gap and to assist with the development of resources for health and social service providers, organizations, and policy makers, we conducted a systematic scoping review to identify and describe the landscape of culturally safe strategies and recommendations for health care and social service encounters with Indigenous families who have experienced family violence. 

## 2. Materials and Methods

We addressed our research objective using systematic scoping review methodology, which is a synthesis method that allows researchers to determine the breadth of the literature for a specific phenomenon and integrate the available evidence into a practical report. Scoping reviews are particularly useful for emerging areas of understanding and as a method to understand the full breadth of the available literature, clarify key concepts, and identifying gaps in research [35,36]. Systematic scoping reviews blend the strengths of the standard scoping review methodology detailed by Arksey and O’Malley [37] (e.g., broad research question, inclusion of qualitative and quantitative evidence, and consultation with key stakeholders to verify findings) with the literature identification methods of systematic reviews to increase transparency and the possibility of replication. To this end, our review was conducted and reported in alignment with the Preferred Reporting Items for Systematic Reviews and Meta-Analyses (PRISMA) Checklist [38]. For our review, a protocol was prepared that followed the six stages outlined by Arksey and O’Malley [37], including the generation of our review questions, which were the following: How is cultural safety in health services for Indigenous peoples who have experienced family violence defined in the literature? What are the principles, practices, and approaches that create culturally safe health care encounters for Indigenous peoples who have experienced family violence? What follows is a brief overview of the activities and procedures involved in the remaining five stages of our systematic scoping review, including literature identification, indexing the data, collating, narratively reporting the findings, and stakeholder consultation. 

### 2.1. Identification of Pertinent Literature 

A multipronged search and identification strategy was developed to collate relevant literature. An electronic database search strategy containing key concepts, terms, and keywords was developed in consultation with an academic librarian and the PRESS Guidelines [39]. Using that search strategy, the following electronic databases were searched for relevant sources on 10 June and 15 June 2020: MEDLINE, CINAHL, PsycINFO, Embase, EBSCO America: History & Life, and EBSCO Bibliography of Native North Americans. Search hits were allowed to span sources included in the databases from database inception until the search date.

The included search terms for the database searches were derived from three concepts: “Indigenous”, “cultural safety”, and “family violence”. These search terms were expanded using synonyms, related terms, and Boolean operators. University of Alberta “Indigenous Peoples (Canada)” search filters available for the specific databases were used to supplement the search terms for “Indigenous”, for example, by including specific Indigenous or Tribal names [40]. The search terms were further expanded upon using subject headings and keywords specific to each database. An example list of search terms for specific databases can be found in Appendix A. Following the database searches, relevant Indigenous journals not found in the databases were hand-searched for eligible articles from inception until the search date, from 18 June to 30 June 2020. A list of Indigenous journals and related search terms can be found in Appendix B. Forward (i.e., screening papers who have cited our included studies) and backward (i.e., reference checking) citation chaining of included sources were completed to capture articles that may have been missed by the database searches and the hand review of relevant journals.

Retrieved sources from the database searches and hand searches were imported into Covidence, which is a web-based workflow platform to facilitate evidence reviews. Two screeners (A.F.; Research Assistant (K.S.)) independently assessed all titles and abstracts, in duplicate, using Covidence’s blind review functionality. Discrepancies on screening were resolved via third-party independent screening (Research Assistant (S.S.)). Independent, full-text review was then completed by three screeners (Research Assistant (S.S.); A.F.; I.A.) and discrepancies were resolved via consensus-based discussion by at least two screeners. 

Articles were considered for inclusion if the article included recommendations for family violence and if it used the key concept of cultural safety *or* if the majority (50% or more) of their sample was comprised of Indigenous participants. This inclusion criterion was selected to both acknowledge the range of accepted terminology in the cultural safety space [33] and also maintain the spirit of cultural safety as proposed by Ramsden [32], and further defined by Curtis and colleagues [34], which indicates that cultural safety should be defined by the Indigenous patient or client rather than by the researchers. 

Additional inclusion criteria were the following: The article provides recommendations for cultural safety in family violence interactions or service provision in a health care context within the results of the paper, or in the discussion directly related to results.Or if the terms “cultural safety” or “culturally safe” are not used in the results, the article includes recommendations in the results section for Indigenous family violence interventions or service provision from a majority of Indigenous participants.The article draws on primary data.The article is not focused on the legal aspect of family violence.The article is available in English.The article is not a dissertation, thesis, book chapter, or personal communication.The article is based on primary data and is not a review, discussion, or editorial.Full-text articles are accessible through the university library system.

### 2.2. Data Indexing

Articles were reviewed based on the eligibility criteria outlined above. A total of 28 articles were included in the initial set. On 30 June 2022, forward citation chaining for each of the articles included in the ‘initial set of articles’ was completed by one author (I.A.) using Google Scholar. Data for each of the included sources were independently indexed by three team members (A.F.; I.A.; Research Assistant (S.S.)) into a pre-populated form in the Covidence program (June 2020) or Excel (June 2022). Indexed information included the following: title, author, author’s Indigenous identity, year of publication, corresponding author’s email address, origin, aims/purpose, research design, community engagement, proportion of participants identified as Indigenous, population, age range of participants, geographic location, Nation/Tribal name, setting, type of family violence, sample size, and demographic requirements of the sample population. Data relevant to addressing the research questions were then extracted by two team members (I.A.; Research Assistant (S.S.)) from the results and discussion sections of the included sources and placed into Excel for analysis and synthesis. Extracted data included the articles’ definition of cultural safety and findings related to the provision of culturally safe care to Indigenous individuals and families who had experience with family violence. 

### 2.3. Collating and Reporting the Findings

Collating and reporting the findings of our review involved three activities. First, we conducted a numerical analysis to capture the number of studies addressing our research question, as well as the proportion of studies (a) located in different regions of the world, (b) focusing on different Indigenous populations, (c) using qualitative versus quantitative research methods, (d) focusing on different types of family violence, and (e) focusing on various gender identities. The second component of our analysis involved a narrative synthesis generated through iterative readings of extracted data related to definitions of cultural safety within the included studies, as well as each study’s findings and associated recommendations related to cultural safety in health care service provision for Indigenous families affected by family violence. Extracted data underwent line by line coding to identify key concepts and constructs related to our research questions; these key concepts and constructs were then grouped within key themes (practically considered as “principles”) for culturally safe care. Analytic memos prepared during the review of the data were used to bolster and track conceptual and thematic thinking. Third, the systematic scoping review findings, inclusive of the numerical and narrative synthesis described above, were presented via a one-hour consultation meeting with members of a local Indigenous community who have a longstanding research partnership with one of the investigators on the research team (C.W.). The purpose of the consultation focused on the appropriate framing and key considerations related to discussing the review’s findings and contextualizing the review’s recommendations. The numerical and narrative syntheses and the consultation exercise are presented in the results section. 

## 3. Results

As noted in our PRISMA diagram (Figure 1), a total of 3783 sources were identified in June 2020, including 1313 articles found through forward and backward citation chaining and hand searching of key journals. These sources were imported into the Covidence system, and 1190 duplicates were removed, leaving 2593 sources to be assessed for eligibility via title and abstract screening. Following conflict resolution by the third reviewer, 314 articles underwent full-text review. A total of 286 sources were excluded at the full-text stage, and 28 articles were retained. On 30 June 2022, forward citation chaining for initial article set (n = 28) via Google Scholar identified an additional 304 articles published between 2020 and 2022, which then underwent screening. After 46 duplicates were removed, 258 of those articles were screened by title and abstract, and of those 15 were retained for full-text review. This process yielded six additional studies for data extraction and synthesis. Thus, a total of 34 studies were included in this review and are included in the synthesis below. 

### 3.1. Article Characteristics

The 34 included articles were published between 1991 and 2022. The majority (35%) of these articles originated from Australia (n = 12); an additional 29% were from Canada (n = 10), 29% were from the United States (n = 10), and 6% (n = 2) were from Aotearoa/New Zealand. The Indigenous participants in these studies represented Torres Strait and Aboriginal peoples of Australia, First Nations, Metis, and Inuit peoples of Canada, Native American/American Indian and Indigenous Hawaiian peoples of the United States (US), Māori of Aotearoa/New Zealand, as well as Indigenous peoples from small island states in Oceania living outside their traditional territories in either Hawaii or Aotearoa/New Zealand [41,42,43]. We acknowledge that these names are largely those assigned by colonizing powers and not necessarily the names that Indigenous communities use for themselves. While this review aimed to reflect Indigenous voices from within the literature as much as possible, and thus set out to report on Indigenous authorship and community involvement as part of the research findings, only 11 articles explicitly indicated having at least one Indigenous author. Many articles (68%; n = 23) did not report on whether Indigenous authors were included.

Most of the studies included some information about how they engaged with Indigenous communities prior to and during the research project, with two studies using Participatory Action Research methods [44,45] and five studies using Community-Based Participatory Research methods [41,42,46,47,48]. Other engagement methods included receiving ethics clearance from local Indigenous ethics boards, having Indigenous community members on the project advisory committees, or spending extended time in the field prior to commencing research. Eleven articles used either “cultural safety” or “culturally safe” in their results section or in the discussion related to the results. The majority of studies (n = 20) included 100% Indigenous participation and perspectives; 79% of the included articles were qualitative studies (n = 27), whereas six (18%) were mixed methods, and one (3%) was quantitative. With respect to types of family violence, the majority (56%) of the included articles focused on IPV (n = 19), with a subset of these papers focusing on perpetrators of IPV (n = 3). Six studies (18%) focused on CM (n = 6), with four of these six articles focused on adult survivors of childhood sexual abuse. The remaining nine articles (26%) included individuals or families who had experienced multiple forms of family violence, including sexual violence. 

### 3.2. Defining Cultural Safety within the Context of Family Violence

Iterative reviews of the included articles reinforced our initial understanding that the concept or phenomenon of cultural safety is inconsistently defined in the literature and many terms are used to describe optimal care for Indigenous clients. Although 15 articles mentioned either “cultural safety” or “culturally safe” care, only 11 of those referenced the concept within their results or discussion, and of those, only 5 articles explicitly framed their study within the context of cultural safety [49,50,51,52,53]. Two of these articles [51,52] drew on the definition of cultural safety from Bainbridge and colleagues [54], who define cultural safety as the practice of “countering tendencies in health care that create cultural risk (or unsafety) through small actions and gestures, not usually standardized as policy and procedure” (p. 43). Dhunna and colleagues [50] framed their use of cultural safety in the context of addressing Māori health disparities, citing the definition by Papps and Ramsden [31]. Funston [49], on the other hand, did not cite a definition of cultural safety, instead providing descriptions of culturally safe services based on participant feedback. For example, Funston [49] states: 


*Cultural safety is not static or definitive but is rather a dynamic and flexible process. Cultural safety relies on services establishing meaningful, accountable, and equitable long-term relationships with communities built on an understanding of their cultures and worldviews as well as their unique needs and strengths. … Moving beyond the limited notion of cultural competency, cultural safety directs service providers to engage in a process of critical reflection. It also means developing a skilled Aboriginal and culturally safe non-Aboriginal sexual assault workforce.*
(p. 3831)

Among the included studies, many times, multiple terms were used within the same article to describe the desired goals and outcomes of health or social services. For example, while Prentice and colleagues [46] referred to cultural safety in their results and provided a definition based on Funston’s [49] work, the concept most often referred to in their article was “culturally appropriate” care. Other articles likewise referred to both cultural safety and culturally appropriate care [55,56], while others referenced cultural safety and cultural competency [57,58]. Other terms included in the articles encompassed the terms of cultural competency [59,60], culturally responsive care [44], culturally appropriate care [41,42,43,47], culturally sensitive care [61,62], culturally- and/or trauma-informed care [45,56], holistic care [63,64], resilience [65,66], and others. Interestingly, all five of the studies that were explicitly focused on cultural safety included either Māori or Aboriginal and Torres Strait Islander participants from Aotearoa/New Zealand or Australia, respectively. This may reflect the origin of the term in that region, and therefore, greater familiarity with and adoption of cultural safety terminology in that region. Collectively, the available global evidence indicates that many terms are used to describe optimal care for Indigenous clients, as described by the Indigenous clients and providers themselves.

### 3.3. Narrative Findings

An iterative review of the included studies informed the generation of three overarching themes that capture recommendations for approaches to cultural safety in the context of working with Indigenous families who have experienced family violence. These themes and their framing as recommendations were reviewed and discussed with Indigenous community members from the Six Nations of the Grand River Youth Mental Wellness Committee on 21 March 2022. Feedback and reflections from Committee Members indicated that these findings resonated with members, however, it was noted that Nations have diverse histories, geographies, ancestral knowledge, and cultural practices. For example, recommendations from Indigenous communities in different regions across Canada may or may not have resonance with a particular First Nations community in Canada, or urban First Nations persons. Committee Members strongly cautioned against portraying or implying a pan-Indigenous perspective, which guided the written narrative of the themes and recommendations presented below. Readers are likewise reminded not to assume that the findings or recommendations from any one Indigenous person or group described below can speak for any other, and yet, readers are also asked to hold the understanding that the forces of colonization have had similar goals, mechanisms, and outcomes across diverse geographical contexts. With this in mind, we sought to reference specific recommendations from diverse Indigenous communities within the article, while extracting high-level themes where repetition was found across articles. 

Three primary themes were generated in our analysis: (1) creating the conditions for cultural safety; (2) healing for people and communities; and (3) system-level change. Importantly, each of the themes presented below work in partnership with each other and none should be viewed in isolation from the others. 

#### 3.3.1. Creating the Conditions for Culturally Safety

Creating the conditions for cultural safety in health and social service encounters with Indigenous families exposed to family violence encompassed several key sub-themes: (a) centering Indigenous peoples and perspectives; (b) recognizing and reducing current barriers to culturally safe care; (c) acknowledging harms and (re)building trust; and (d) cultivating unhurried, non-judgmental relationships.

##### Centering Indigenous Peoples and Perspectives

Studies were consistent in their findings and reflections that Indigenous clients felt safer to engage in disclosure and healing when they felt listened to and could see themselves centered and reflected in the resource, service, or organization they accessed. Understanding and incorporating Indigenous worldviews, values, and cultural contexts helped clients feel understood and contributed to a lower likelihood of experiencing stereotyping and stigma [42,48,49,52,57,58,67,68]. The call for non-Indigenous providers to listen and to center Indigenous peoples and perspectives in family violence recognition and response was captured by Funston [49]: 


*Participants recommended that ‘mainstream’ services meaningfully incorporate Aboriginal and Torres Strait Islander Worldviews into service delivery. Aboriginal participants described their worldviews as ‘central’ to developing cultural safety. One Aboriginal participant expressed this by stating, “we need white people listening and to take on Aboriginal principles, then it would be safe to do the healing”.*
(p. 3825)

Authors and participants of the included studies expressed a need for communities, service providers, and institutions to first understand the reality of family violence from the Indigenous person’s and community’s perspective. Counter to Western theories of family violence within many of the studies, family violence was described by Indigenous participants as a complex social and historical phenomenon, enveloping whole families and their communities through multiple generations, and as a product of cultural dispossession, historical violence, and specific colonial and mainstream racist policies that sought to separate and weaken Indigenous families [43,46,47,64,67,69]. Authors and study participants juxtaposed this historically contingent and structural perspective against commonly held Western theories and service models that tend to frame and treat instances of family violence as discrete, individually based, and temporally bound phenomena [48,64]. Family violence was described in one article [56] as inextricably linked with experiences of mental health issues, substance abuse, and poverty, with each of these outcomes constituting one part of a “constellation of traumas” (p. 44) stemming from colonial policies of racism and forced family separation, including residential schools [48,56,61,65,68]. Family violence was also seen to stem from the intergenerational cycles of abuse that were borne from the disrupted generational transmission of culture [49,53,55,56,59,61,70] and which led to the loss of traditional gender roles [55,61,64,70]. 

Exemplifying the heterogeneity of Indigenous worldviews included in the available literature, Rankine and colleagues [43] described diverse ontological explanations for the causes of family violence among a group of participants from various Oceanic communities. Bagwell-Gray and colleagues [62] also acknowledged the variation in experience and perspective between Indigenous communities in the US, highlighting the pragmatic tensions between developing a multitude of community-specific and culturally specific programs or investing time and resources in a cultural safety intervention that is broad and encompassing enough to help many Indigenous communities affected by family violence. Beyond the challenges of implementation and scale, there was consistency across the studies that Western theories and approaches, such as the Duluth Power and Control Wheel, were insufficient for understanding and addressing family violence in an Indigenous context, as they do not reflect an Indigenous ontology that includes a relational network of family, kin, community, spirituality, and traditional customs [43,71].

Studies recommended healing services and interventions to be designed in partnership with Indigenous communities and founded on an Indigenous worldview of healing and health [47,49,57], centering on culture, spirituality, and relationships with the land. Integrating common Western approaches, such as cognitive-based therapy (CBT) and trauma-informed services were viewed as successful only when integrated *into* the Indigenous-led organizations and/or frameworks [55,56,59,60]. 

##### Recognizing and Reducing Current Barriers to Culturally Safe Care

The harm related to barriers Indigenous families face in accessing culturally safe care was identified among the negative effects experienced because of family violence. These past and present-day barriers were said to further break down Indigenous family systems and lead people to avoid accessing support when they need it. For example, common experiences of racism, victim blaming, stigmatization, judgment, and stereotyping contributed to a lack of trust and a climate of fear in the anticipation, or the actual experience, of encountering non-Indigenous providers and service organizations [50,51,52,53,61,68,69]. Culturally inappropriate, racist, and stigmatizing ‘care’ experiences resulted in an unsafe environment [71] and were viewed as increasing the risk for additional harm when seeking support for family violence-related concerns [50,72]. Indigenous participants in the included studies felt excluded from, and unsafe accessing, services that were perceived as culturally inappropriate or insensitive [71,72], used an exclusively Western worldview of family violence [48,49], failed to offer the service in the Indigenous person’s language [57], or were based on a pervasive attitude of tolerance or acceptance of family violence in Indigenous communities and a lack of accountability for perpetrators [63]. In rural and remote contexts there were often few or no community-based shelter and treatment services available [57] and limited access to Indigenous providers [46]. Poor availability of transportation, technological infrastructure, and a lack of financial resources prevented Indigenous families’ access to available services that were many hours away [52,57,68]. The impacts of these barriers contributed to the increased use of informal support systems, including support from family, friends, and informal helpers within the community [47,57,70]:


*Many women do not report family violence as being a part of their daily life, as they see no point in asking for assistance where there is none to be had. The standard response of physically removing women from the community to address issues of safety and medical care, … is often seen as more traumatic, as families and communities are torn apart, a practice that is in direct conflict with the [Inuit] worldview …*
[57] (p. 10)

##### Acknowledging Harms and (Re)building Trust

Both the feeling of cultural safety and the decision to disclose family violence were consistently linked to a relationship of trust with the provider and organization [42,47,48,51,57,58]; as aptly stated by a male-identified focus group participant in one of the included studies, “without that trust, without that bond, without that belief … if that’s not there, they’re just going to go this is not worth me copping another beating when my partner finds out about it” [53] (p. 217). Non-Indigenous providers were encouraged to understand and acknowledge the legitimate fear and distrust that may be experienced by Indigenous clients [69]. The intense fear of mainstream service providers and organizations and a lack of trust due to historic and current harms led participants to be wary of disclosure and to actively assess signs of cultural or physical safety when considering a resource, provider, or organization [43,49,50,51,53,57]. 

While some authors reported participant reliance on their kinship networks to identify safe services (see “Borrowed Trust” in [51]), other authors described the ways that participants sought out signposts from the provider or organization to signify the presence of cultural safety in a resource or service. Indeed, some Indigenous participants requested that providers and organizations *demonstrate* they could be trusted and were culturally safe, with one person stating: “We need to create accountability from the organisations that will provide the help … It’s about putting the accountability back on the resource” [53] (p. 216). Recommendations for communicating this commitment included, for example, a verbally recorded statement from program/organizational leadership; the involvement of the Indigenous communities in the development of the program or resource; and the presence of colors, artwork, and symbology from Indigenous cultures [53], recognizing that these would differ depending on the culture. For many Indigenous clients, seeing themselves presented in the service, including their language, worldview, cultural elements, and other people that looked like them, helped indicate that a provider or service was culturally safe and provided a sense of belonging [51,61,69]: 


*In an abuse support group where “three women had brown eyes like I did”, one woman believed that she “wouldn’t have got in touch with what I did had it not been for the other native women. There was no searching, no scrambling. I did not have to seek out to belong”.*
[69] (p. 230)


*The men wouldn’t come if it wasn’t culturally appropriate, and I think that’s true.*
[71] (p. 67)

Importantly, many of the included studies reported that participants felt most comfortable talking to Indigenous counselors, or working with Indigenous organizations, who already understood their culture, approaches, and background [42,43,44,52,53,61,64]. In one study [62], participants expressed that service providers must understand that their communitarian values are foundational to their well-being and identity, making “the decision to leave an abusive relationship … untenable when doing so would be synonymous with cutting those [community-wide and caregiving] ties” (p. 169). In other studies, participants identified the importance of providers who understand the Indigenous client’s language and use the appropriate way of speaking about sensitive issues [45], as well as language that reflects their community’s cultural gender norms [43].

In the two articles that focused on services for children and families involved with child welfare agencies, a foundation of Indigenous culture was perceived as central to the improvement of child and family well-being [44,60]. Parents who had experience working with a collaborative and trauma-informed family preservation model for urban Indigenous families in the US explained the following: 


*It feels comfortable there because you feel the culture there. You see it all around, and then you feel it come through the teachers as well. And there’s certain topics and issues that we talk about that we all can relate to because we’re the same culture.*
[44] (p. 106)

Providers and their organizations were also encouraged to allow clients to incorporate trusted friends, family, and community members into, for example, IPV-related intervention or shelter settings; their inclusion was believed to improve the feeling of safety where one might not otherwise exist [48,67]. Participants also suggested that service providers acknowledge past harms clients may have experienced in the service system and offer to meet in informal spaces that feel safer to Indigenous clients, rather than meeting at mainstream organizations with unsafe histories [49,68]. 

##### Cultivating Unhurried, Non-Judgmental Relationships

A non-judgmental environment characterized by an unhurried approach to patient or client care was also central to many participants’ decisions to disclose their family violence experiences. An unhurried approach placed the discussion pacing and timing into the client’s control and took time to build the relationship with the person before asking about exposure to family violence [51,52]. The experience of care in an unhurried environment is captured by the words of one participant: 


*You’re not like a number. You’re a person and they go out of their way to make sure everything’s done really thorough. I’ve just seen the doctor here for five minutes. At the AMS [Aboriginal Medical Service], I would have seen my doctor for probably an hour and a half. And everyone’s like that. It’s not like anyone rushes there. Everyone calls it Koori time.*
[52] (p. 802)

Having time to establish rapport with a provider who understands “how Aboriginal Families work” [72] (p. 10140) was seen as essential to creating trust and a feeling of safe space where one would not be judged. In one study [55], a traditional healer used the term “ease” to capture the preferred pace and sensitive nature of counseling. Others spoke to the value of recognizing the stigma and shame that clients may have experienced and ensuring that their dignity and worth were honored via patient, non-judgmental responses to disclosures of violence, particularly disclosures of violence perpetration. This is captured by the following quote: 


*I try to make that person feel that he is just one of billions—or thousands of people and there is help—and just like anything else it can be treated. I believe in a spiritual way it can be treated.*
[61] (p. 29)

#### 3.3.2. Healing for Individuals and Communities

Among the included articles, interventions for family violence were rarely discussed solely in terms of helping an individual find safety or leave a violent situation. Rather, the response to family violence was described through the lens of holistic healing at the individual and community level, which involved a deepening of connections to self, community, and culture, and restoration of Indigenous identity and well-being. Throughout many of the articles, healing was described as a journey [48,53,56,61,65] that required interventions that attend to all levels, including the individual, family, and community [43,48,57,63,64,65,67], as well as holistic treatment of the issues co-occurring with family violence including mental health, substance abuse, and other trauma-related concerns [50,56,71,73].

For some, the journey towards healing included naming and addressing the harms that were inflicted on their families and communities through colonial policies and continued in mainstream practices [53,54,56,69], as well as naming current experiences of harm, including the cyclical nature of victimization and abuse [56]. Within healing programs, providers were encouraged to incorporate education about colonization and intergenerational trauma into healing services [55,56,68]. In one article, it was important for the community to engage with the meaning of family violence as part of their healing: “For this community, the first step in addressing family violence [was] to ensure that everyone participate[d] in the community’s naming and conceptualization of it” [64] (p. 55).

Healing approaches included disentangling the influence of Western colonial practices and values on Indigenous communities, while reclaiming important aspects of pre-contact Indigenous culture, including, for example, matriarchal community structures and peaceful relationality among family, kin, and community. For example, for men, talking about the role of colonization and Western culture in family violence patterns helped clients to become conscious of destructive, patriarchal gender roles and to “… realize it hasn’t always been this way. Five hundred years ago it wasn’t like this … this is a consequence of colonization. This gender role is not a traditional role for a man” [55] (p. 46). Making the connections between colonization, patriarchy, hegemonic masculinity, and gender-based violence helped some clients see their Indigenous values and traditional roles as potential sources of healing, rather than the cause of the violence [55,61]. Challenging the notion that violence is inherent to Indigenous culture as well as reclaiming the power of traditional culture and gender roles were identified as protective and healing factors for both male perpetrators [71] and women who experienced IPV [62,66].

In many of the articles, healing that related to reclaiming one’s identity involved strengthening connections to family, community, culture, and land and restoring spirituality and traditional practices [50,53,55,56,57,58,59]. Applied examples from participants and providers included, for instance, the inclusion of traditional practices in healing services [47,48,53], support reconnecting with lost children, (re)gaining Indigenous status [69], and engaging with interventions that provide strategies to stay connected to safe Indigenous community members, Indigenous knowledge, and traditional land, especially for those who have had to leave or live outside the community [62]. In a study with native Hawaiian women who had experienced IPV [47], the author recounted the following:


*… reconnecting with a cultural base through traditional cultural practices of working in the lo’I (irrigated terrace for taro), chanting, and hula helped participants reclaim their identity, [one participant said] “it got me to realize who I am, you know, the way I should be, and the way I’m supposed to be. … And it helps me to realign myself back to where I should be in my life”.*
(p. 7)

Studies made a strong case for the involvement of family members in healing, as well as the inclusion of Elders as role models and guides [48,55,57,58,61,67]. Moreover, storytelling, which was framed by some as a culturally relevant way to speak the truth of individual and collective experiences, was also identified as an important conduit of healing [41,48,56,61,64,67,69]. Naming violence and having the opportunity to share personal stories in groups with other Indigenous peoples gave both survivors and those who harm the opportunity to see that they were not alone [48,61,64]. Using other traditional ceremonies and practices, such as healing circles and sweat lodges, offered a safe place to voice traumas [56,61], providing a “spiritual framework for addressing problems … and heightened the participants’ motivation for confronting difficult issues” [67] (p. 8). Sharing stories in a well-guided workshop also facilitated cultural learning and sharing of Indigenous knowledge between participants, “… a process that shifts power from the dominant culture and recognizes the value of Anagu [Indigenous Aboriginal] knowledge in understanding and responding to contemporary challenges” [45] (p. 381). 

Importantly, several articles offered caveats and counter-perspectives regarding the blanket involvement of family and Elders in responses to family violence, suggesting that a thoughtful and cautious approach is necessary. Including Elders without careful understanding of the complex community power dynamics may be “overly simplistic”, as not all Elders may hold authority within communities, and some Elders may themselves have used violence towards family members [49] (p. 3824). Likewise, studies raised concern about the limits to confidentiality and the risk of harm when disclosing violence to Indigenous providers within a small community context [42,47,48,53]. Some identified this as a need for many more providers to be trained [46,48], and others reported that mainstream services may be preferred by some Indigenous peoples [53,72].

Studies were also careful to point out that providers need to temper any assumptions regarding a client’s desire to engage with their community or culture in healing from family violence, and that engagement with culture and community is something that should be driven by the client in accordance with their own needs and preferences [61,67,69]. Similarly, in circumstances where children have been placed in out-of-family care related to experiences of family violence, studies cautioned providers about the blanket involvement of family. Rather, it was noted that “kinship placements could be protective for some children, whereas for others, family placements actually exacerbated behavioural problems” and that “one size did not fit all” for children in care [60] (p. 901). 

#### 3.3.3. System-Level Change

Importantly, recommendations within the included literature went beyond individual practitioner and organizational practices and approaches to include changes in policy and practice at the system level. Examples of system-level changes included provider training with a focus on self-knowledge, ensuring equity of power and funding to Indigenous peoples and priorities, and government-level changes to create culturally safe interventions and care. 

##### Provider Training with a Focus on Self-Knowledge

Strengthening Indigenous workforces and collaboration between non-Indigenous and Indigenous providers or organizations was a widely recommended approach to facilitate connection to safety, culture, and traditional practices [47] (see for example [48,49,55,56,58,59,60]). Authors emphasized the need for providers to understand today’s social and health challenges in the context of historical events and harms [55,56,61,70], and recommended that all non-Indigenous providers engage in formal, compulsory training on the impacts of collective traumas stemming from colonization, structural racism, and continued white privilege [46,48,49]. It was also recommended that all training attend to the unique regional history, and specific individual or community experience of violence [42,56,61]. A few articles also included recommendations for additional provider training on family violence signs and impacts [42,57,63] and on trauma-informed care [60]. 

Two studies recommended that Indigenous peoples who are training to become family violence service providers should also complete curriculum on the history of colonization, forced removal of Indigenous children, and dispossession of Indigenous peoples from their lands [49,74], as this history cannot be assumed to be known due to barriers in access to community knowledge, including intergenerational shame. Authors also suggested that Indigenous providers receive additional social, emotional, and financial support during training [74], as well as supervision, ongoing training, and job support once they are in practice [46]. 

##### Ensuring Equity of Power and Funding to Indigenous Peoples and Priorities

Long-term relationship building between government agencies and Indigenous communities was described as essential for safely responding to all forms of family violence [44,46,49]. In work by Lindeman and colleagues [45], workshops that drew on pre-existing relationships built over time between Indigenous and non-Indigenous practitioners, staff, researchers, and senior community members were identified as “important given the sensitive nature of the topic [sexual violence] being explored” (p. 381). Participants in the included studies requested that mainstream organizations and non-Indigenous service providers spend time building relationships with the community [46], including meaningfully engaging with Indigenous peoples outside of professional or formal contexts, and ‘yarning’ (i.e., building relationships over time through informal, non-directive conversation) with communities for at least 12 months prior to establishing family violence services [49]. Short-term contracts and inconsistent funding for these services were identified as significant barriers to establishing the trust that was deemed necessary to cultivate meaningful and sustainable relationships with communities [46,49]. 

##### Organizational and Government-Level Changes

System-level changes at the organizational and government level were identified as necessary supports to bolster individual practitioner efforts towards creating cultural safety [46,60]. Inter-agency collaboration and communication could break down access barriers such as transportation and childcare for clients [63,68], as well as serial experiences of stigmatization and racism, that perpetuate unsafety and violence. For example:


*Challenging the credibility of these young Maori mothers was not just practiced by their abusive partners but also frontline personnel in government such as WINZ [work and income support] and HNZ [Housing support]. For instance, when Hana approached HNZ, the individual at the front desk used Hana’s mother’s gang violence history to blame her for being pregnant and ‘using up’ HNZ’s services. Participants were caught in a web of intergenerationally based institutional discrimination.*
[50] (p. 20)

Deeper changes to the organization of family violence services were recommended to decolonize the broader system that providers operate in and to provide services that are organized around Indigenous community priorities. Within a subset of articles, for example, recommendations included a re-imagining of service interventions to include support for both victims and abusers together [49,64]. An approach that includes both victims’ and abusers’ voices and healing was felt to increase accountability [71] and assist women to remain in place in their community [62]. Co-offered services were also seen as early intervention strategies for victims, as those who harm have also often been victims of violence, with the understanding that victims are more likely to become those who harm or use violence in the future [49]. 

Additionally, recommended changes to the organization of family violence services included expanded notions of intervention and healing that included increased education and outreach to Indigenous communities. Outreach was recommended to improve the community’s understanding of family violence and to support the informal and formal community-based prevention and intervention efforts. Participants asked for community outreach, targeting education on family violence for youth [50,57,60,63,64], improved knowledge of prevention and support [41,44,46,66,70], and reduced community shame [49,64]. Community development opportunities tied to family violence services, such as employment support and recreation infrastructure [44,47,68], or social programs such as cooking classes that promote community well-being and extend the sense of community post-intervention [48], were also recommended. 

Studies recommended that the whole government and policy action should be based on a historically nuanced understanding of family violence and health inequity in Indigenous communities; this nuanced understanding would fully appreciate the contextualized legacy of colonial harms. Having this understanding would canonize an alignment between policy actions and Indigenous priorities and frameworks, and meaningfully include the provision of resources for ongoing collaboration and funding to develop, deliver, and sustain family violence education and intervention programs that meet the needs of the community, as defined by the Indigenous communities themselves [44,49,50,64]. For example:


*Participants felt that the Western theories currently governing the child welfare system are both ‘inappropriate and colonial’. Aboriginal participants emphasized that in order to decolonize services and create cultural safety within the service system, it is necessary to view violence as a product of the socio-political context … [and thus] dismantle the racist and ahistorical believe that violence is a product of Aboriginal and Torres Strait Islander cultures.*
[49] (p. 3824)

Similarly, participants asked for policies and protocols to be put in place that explicitly acknowledged past harms and validate current fears [44,49,50]. For example, specific policies and protocols between child welfare departments, health services, and Indigenous communities were requested [44] to account for the intense “double jeopardy faced by Aboriginal women when they seek safety for themselves and their children, but fear removal of their children if they disclose violence” [52] (p. 802). Likewise, in a Māori context, Dhunna and colleagues [50] pointed out that Māori frameworks already exist within the Aotearoa/New Zealand government, but must be executed and equally funded to appropriately redress the colonial legacies of violence and dispossession. Engaged and sustained consultation with Indigenous communities would help policy makers and organizations better understand the Indigenous communities’ experiences and support the existing community-led programs and methods of healing that are already trusted and accessed. By doing so, consultations could meaningfully shape health care policy and delivery, “make strong advances in the reduction of shame and stigma”, and ultimately contribute towards the goals of reconciliation [57] (p. 12).

Underlying the totality of these changes were some participants’ desires for mainstream society to change its relationship with, and perception of, Indigenous peoples. Highlighted approaches included electing Indigenous representatives to local government, appointing people to the hospital and community leadership boards, and sharing Indigenous history, arts and crafts, and Elders’ stories [64] (p. 23). 

## 4. Discussion

Our findings highlighted practices and approaches to culturally safe family violence interventions that require ongoing provider-, organizational-, and government-level engagement with self-reflection and accountability to examine and address the policies, theories, attitudes, and practices that have contributed and led to family violence in Indigenous communities. The three themes generated from our analysis (creating the conditions for cultural safety; healing for individuals and communities; and system-level change) coalesce around the goal of establishing health equity through a recognition of historical and current-day barriers to healing, as well as a transfer of power to Indigenous clients and communities through culturally safe practices, approaches, and policies. Our search inclusion criteria encompassed studies based on primary research that explicitly focused on cultural safety in family violence interventions, or that included recommendations for family violence interventions drawn from many Indigenous participants, whether service users or providers. The latter criteria is in alignment with the concept of cultural safety described by Ramsden and Papps [31,32] and defined by Curtis and colleagues [34]. 

Recommendations from our review illustrate that Indigenous peoples’ experiences and knowledge of family violence are distinct from typical Western theories and interventions. Culturally safe interventions follow from an Indigenous understanding of family violence and of healing, including the healing needs of individuals, families, and communities. At the service provider, organization, and government levels, an urgent need to repair trust and relationships between Indigenous families and family violence services in a broader, historically contingent way was identified. Distrust and fear of family violence services was widespread and warranted. The history of racism and stigma is still present in the policies and practices of family violence services today [22], demonstrated by the continued over-representation of Indigenous children in child welfare services [9,10,14] and the dearth of historically informed and culturally safe interventions for Indigenous families [75]. The importance of taking time to build relationships of trust, meeting experiences of violence with non-judgment, and slowing the pace of the health or social service encounter are central to cultural safety, echoing the findings from related reviews [22,76].

Consistent with the existing literature, our systematic scoping review echoes a bias in the literature toward secondary and tertiary preventative interventions that aim to address the consequences of violence exposure and/or reduce their recurrence. There remains a dearth of evidence-based recommendations for primary interventions to *prevent* family violence in Indigenous communities [77]. Despite this, recommendations from the included studies demonstrate that culturally safe interventions require changes to take place upstream at the policy and system levels. These recommendations were included as a part of our third theme, system-level change, contributing to calls for cohesive policy on culturally safe family violence services [78]. Increased access to Indigenous providers was also recommended in our findings; however, barriers to enrolling, training, and retaining Indigenous health care and social service providers [79] must be addressed to effectively meet this need.

Our results have many parallels with the recommendations offered in a recent review of trauma-and-violence-informed care (TVIC) for Indigenous families seeking family violence services in primary care contexts [80,81]. That review drew on a framework for health equity in Indigenous health care that defines TVIC as responding to colonial- and state-induced traumas and includes cultural safety as a central element of the TVIC framework [82]. Our findings adhere to cultural safety as a primary framework for understanding family violence, as it precisely responds to broader power dynamics experienced by Indigenous peoples living within a post-colonial context, distinct from trans-national or multicultural approaches that can be applied to any group. In other words:


*Indigenous people must be seen not as one cultural or ethnic group amongst many, but an equal founding nation and therefore with a rightful claim to a pre-eminent status.*
(Ramsden, 2002 p. 175, cited as Ramsden, 2004 in [83] (p. 14))

Consistent with Indigenous-led restorative action, a cultural safety framework is consistent with the larger socio-political goals of transforming power dynamics between Indigenous and Western governments, including through funded services. More specifically, cultural safety (as discussed herein) goes beyond any one program or policy initiative to require the whole of government and public efforts to acknowledge and account for historical and ongoing harms perpetuated by colonial and Western governments, programs, and policies. This includes an effort to change the understanding of the origins and continued experiences of family violence in Indigenous communities: change that may be shepherded through by public education and policy-level support. The actions identified in this review, required to provide culturally safe interventions for Indigenous peoples experiencing family violence, are aligned with the goals of such initiatives as the Truth and Reconciliation Commission in Canada [84], the Wengai Tribunals in Aotearoa/New Zealand [85], Reconciliation Australia [86], and in state- and Tribal-wide efforts throughout the US (e.g., Maine-Wabaniki-State Child Welfare TRC [87]). 

### Future Research Directions

Although there were no geographic limitations to our systematic scoping review, the included articles drew dominantly on recommendations and experiences of Indigenous service providers and clients navigating health and social systems in the Global North, including Australia, Aotearoa/New Zealand, Canada, and the United States. Within these contexts, the experience of Indigenous communities vis-à-vis colonization and post-colonial governments has been diverse. Interpretation of our results should therefore be mindful that the specific requirements for cultural safety must be defined by, and in partnership with, the Indigenous community being served, and will be judged by the Indigenous communities themselves. A small number of included studies focused on the experiences of Oceanic peoples living outside their home countries in larger Western nations. Although some excellent Indigenous-led and Indigenous-researched interventions have been discussed elsewhere [88,89], additional research on Oceanic people’s experiences and needs with regard to family violence interventions is needed [90]. Likewise, our review focused heavily on the role of Western governments and colonial policies but did not include a discussion of the role that the Christian Church played in many of the policies and practices that led to and perpetuated violence in Indigenous communities within these same countries and worldwide [84]. The complex evolution of the relationship between Christian and Indigenous spirituality, as well as the role of Church leaders in both perpetuating and healing from violence, would benefit from focused and thoughtful consideration in future reviews of culturally safe approaches to care. 

Though our review identified a small number of studies focused on interventions and perspectives of men and boys, the available services for males as both children and adults, as both victims of violence and those who harm others, represents a gap in the available family violence interventions [49,55,56,64]. Our study excluded articles with a focus on the judicial system, where some of this literature may be found; however, this gap also reflects the relatively early stages of research on culturally safe services for Indigenous male victims and male perpetrators [91]. Additional research on how to best organize, support, and operationalize co-designed victim/perpetrator services in a culturally safe way is also needed. Importantly, included studies also lacked representation or recommendations on culturally safe family violence interventions for Indigenous 2SLGBTQIA+ peoples; this gap represents compounded and intersectional trauma, given the multiple layers of violence experienced by this community [92]. 

Although our review set out to highlight Indigenous voices, whether Indigenous authors were represented as part of the publication as well as their relationality were not consistently reported. Furthermore, our study only included academic sources of literature, and did not include grey literature. As identified, there is a need to address the gaps in Indigenous authored literature on family violence; this is an actionable mechanism for ensuring “sustained investment in efforts to synthesize diverse sources of knowledge, support for open source publications, and structural support for Indigenous control of knowledge collection and dissemination regarding policy development related to their communities” [10]. 

Lastly, research evaluating culturally safe interventions is greatly needed in the family violence literature, and thoughtful, Indigenous-driven evaluation research that respects Indigenous knowledge and cultural practices can be healing and promote health equity [93]. Only three studies integrated or reported on measures of evaluation for their cultural safety intervention [44,59,73], and one study framed their work within the purview of quality improvement [60]. Two studies explored the adaptation of Western interventions to an Indigenous context [61,73], while two others tested Indigenous women’s specific perceptions and experiences of existing Western interventions [51,52]. Given that the outcome of cultural safety and both the nature and goals of healing should be defined by the Indigenous clients and communities, it is recommended that future studies center Indigenous worldviews and research methodology and use the evaluative criteria defined by the community and service users themselves [49,94]. Careful consideration and caution are also warranted, however, as academic evaluation studies can be harmful via the use of valuable resources for evaluation objectives or methods that may be incongruent with community priorities or needs. 

## 5. Conclusions

Cultural safety, cultural competency, and culturally appropriate and related terms have all been variably used within the included literature exploring how best to provide health care and social services to Indigenous individuals, families, and communities affected by family violence. The findings of our systematic scoping review indicated that, in line with the guiding definition provided by Curtis and colleagues [34], the provision of culturally safe care is best considered an outcome of a health care or social service encounter that is experienced and defined as such by an Indigenous individual, family, or community seeking care. Our findings also offered several key and actionable recommendations that can bolster the possibility with which Indigenous community members who have experiences of family violence consider their care interactions as culturally safe. These actionable recommendations encompass creating the conditions for cultural safety via centering Indigenous peoples and their perspectives in defining what cultural safety is and looks like, recognizing and reducing current barriers to culturally safe care (as defined by Indigenous communities); acknowledging the extent and persistence of colonial harms, including its role in the onset and/or persistence of family violence in Indigenous communities; as well as nurturing unhurried relationships characterized by patience, respect, and reciprocity with Indigenous communities. Second, honoring the need for and inextricable links between individual- and community-level healing is central to mobilizing change in the levels of family violence experienced by Indigenous communities. Finally, system-level change in the form of policy and infrastructure that allows for—and requires—provider cultural safety training, as well as equity in power and funding for Indigenous peoples and priorities are critical. 

## Figures and Tables

**Figure 1 ijerph-19-16967-f001:**
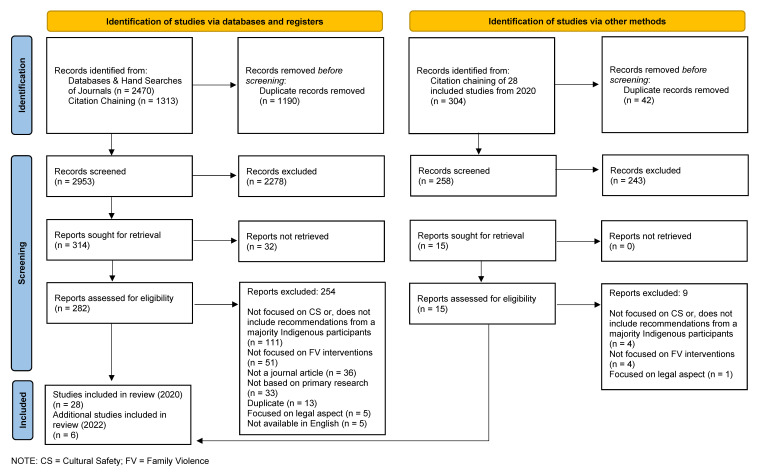
PRISMA Flow Diagram of Included Sources.

## Data Availability

Not applicable.

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
