# Peer review of "Indigenous Cultural Safety in Recognizing and Responding to Family Violence: A Systematic Scoping Review"

_ijerph, 2022, doi:10.3390/ijerph192416967_

Round 1

Reviewer 1 Report

I thought that the article was very strong, well done, and addresses a highly relevant topic related to Indigenous family violence. The discussion and recommendations were well addressed and show a route to future research and practice.

Author Response

Please see the "Reviewer 1" section in the attachment. 

Reviewer 2 Report

Thank you for the opportunity to review the manuscript titled "Indigenous Cultural Safety in Recognizing and Responding to Family Violence: A Systematic Scoping Review." This is a review article that investigates themes across existing research to develop best practices for developing and providing culturally safe social programs for Indigenous peoples experiencing family violence.

The abstract is well-written and informative.  Typo on line 16, should read June 2020.

Introduction:  The first sentence requires a citation, and possibly examples of "serious negative short- and long-term consequences." This paragraph ends with a citation and broad examples of such consequences.  The authors should consider merging these two statements and expanding on what these consequences actually include (e.g., anxiety, depression, PTSD, suicide, etc.).

Throughout this section, the authors refer to "Indigenous peoples" and "Indigenous people."  The authors should be consistent in the use of the singular or plural when using these terms.

Line 85, Page number provided...is this for the quote in the prior sentence?  If so, should move the refernce to that sentence.

Line 98: This sentence seems awkward to me "...inclusive of various terminology systematic as we..."  Is a word missing or should a word be removed?

Line 163: change "was" to "were"

Line 171: remove the comma after "violence"

Line 174: remove the comma after "space[32]"

Line 581: hyphenate "out-of-family", as this sentence is difficult to read without the hyphens.

Line 711: should this read "Indigenous peoples' experiences..."

Overall, this is a very well-written manuscript, and the themes extracted from the authors' scoping review seem both warranted and valid.  The conclusions are both important and practical when designing and providing services for Indigenous peoples experiencing family violence.  I am appreciative of the level of detail the authors provide in their Appendix for how they identified relevant articles for their review.

Author Response

Please see the "Reviewer 2" section in the attachment. 

Reviewer 3 Report

This is a very important and well written review - I was a pleasure to read and will make a significant contribution to the field.

There are some minor issues that require attention.

Some Spelling and grammatical errors require correction (e.g., line 16, 163, 289 - 291), including use of singular and plural forms of concepts (data is the plural form for 'datum') and tenses (ensure past tense is used consistently).

Furthermore, ensure that you explain what is meant by "forward and backward citation chaining"

Author Response

Please see the "Reviewer 3" section in the attachment. 
